# Real World Experience of Second-Line Treatment Strategies after Palbociclib and Letrozole: Overall Survival in Metastatic Hormone Receptor-Positive Human Epidermal Growth Factor Receptor 2-Negative Breast Cancer

**DOI:** 10.3390/cancers15133431

**Published:** 2023-06-30

**Authors:** Ji-Yeon Kim, Junghoon Shin, Jin Seok Ahn, Yeon Hee Park, Young-Hyuck Im

**Affiliations:** 1Division of Hematology-Oncology, Department of Medicine, Samsung Medical Center, Sungkyunkwan University School of Medicine, Seoul 06351, Republic of Korea; 2Department of Health Science & Technology, Samsung Advanced Institute for Health Sciences & Technology, Sungkyunkwan University School of Medicine, Seoul 06351, Republic of Korea

**Keywords:** second line treatment, palbociclib with letrozole, hormone receptor positive, HER-2 negative, metastatic breast cancer

## Abstract

**Simple Summary:**

Second-line treatment strategy after the first-line CDK4/6 inhibitor with aromatase inhibitor is considered by the behavior of hormone receptor positive human epidermal growth factor receptor-2 negative (HR+HER2−) metastatic breast cancer (MBC). Progression free survival 2 was one of the association factors for overall survival of HR+HER2− MBC. Therefore, the second-line treatment strategy was important to improve prognosis in patients with HR+/HER2− MBC.

**Abstract:**

Background: We analyzed real-world practice of second-line treatment in hormone receptor (HR)+ human epidermal growth factor receptor-2 (HER2)− metastatic breast cancer (MBC) following the first-line CDK4/6 inhibitor with letrozole. In addition, we evaluated the relationship between second-line treatment strategies and survival outcome. Methods: Using the clinical data warehouse, clinical information including MBC diagnosis, treatment and survival outcomes were collected. Results: In total, 305 patients were treated with the first-line palbociclib plus letrozole, and we evaluated 166 patients who were treated with second-line treatment. Of the 166 patients, 28.5% were treated with capecitabine (C), followed by exemestane with everolimus (EE) (27.3%) or cytotoxic chemotherapy other than capecitabine (T) (18.8%) and fulvestrant-based treatment or endocrine monotherapy (F) (12.7%). Eighteen patients (10.9%) were enrolled in clinical trials (CT). With regard to treatment strategies, and the median progression-free survival of second-line treatment in a metastatic setting (PFS2) was 7.4 months with C, 5.2 months with EE, 4.8 months with T, 3.6 months with F, and 3.6 months with CT (*p* = 0.066). In patients with visceral organ disease progression, C (31.3%) or T(31.3%) was the most common second-line treatment followed by EE (21.9%). Most of the 47 patients with bone metastasis alone were treated with EE (38.2%), followed by C (23.4%) and F (21.3%) (*p* = 0.008). The median overall survival of second-line treatment in a metastatic setting (OS2) was 42.3 months with C, 35.7 months with F, 30.7 months with EE, and 23.1 months with T. The median OS2 for those in CT was not reached (*p* = 0.064). ER driven BC, disease progression site and PFS2 were associated with OS and OS2 in HR+HER2− MBC (*ps* < 0.05). Conclusions: We suggested the second line treatment strategy was important to improve prognosis in patients with HR+/HER2− MBC, especially given the recent standardization of first-line treatment and the many available second-line options.

## 1. Introduction

The combination of a cyclin-dependent kinase 4 and 6 (CDK4/6) inhibitor with endocrine therapy (ET) is the current standard of care for patients with hormone receptor-positive (HR+) human epidermal growth factor receptor 2-negative (HER2−) metastatic breast cancer (MBC) which is defined by the expression of either the estrogen receptor (ER) or progesterone receptor (PgR) in at least 1% of tumor cells and the absence of HER2 overexpression or amplification [1,2]. There are three CDK4/6 inhibitors: palbociclib, abemaciclib, and ribociclib. Several clinical trials have demonstrated significantly improved progression-free survival (PFS) with the combination of a CDK4/6 inhibitor and an aromatase inhibitor (AI) compared to an AI alone as the first-line treatment of patients with HR+HER2− MBC regardless of menopausal status [3,4,5,6]. In terms of overall survival (OS), the combination of an AI with ribociclib or abemaciclib improved survival outcomes compared to AI treatment alone. However, AI plus palbociclib did not have a statistically significant OS benefit [7,8,9,10]. Recent real-world database analysis demonstrated significantly longer OS with palbociclib plus AI compared to AI alone as the first-line treatment for HR+/HER2− MBC, but no OS benefit in the PALOMA-2 clinical trial caused by palbociclib could not be included in category 1A recommendation in the current practice guidelines [2,11,12].

In second-line treatment, endocrine therapy (ET) with or without targeted agents is recommended as the second-line treatment after CDK4/6 inhibitor use, unless the patient is refractory to ET according to the treatment guidelines for HR+HER2− MBC [2,12]. However, despite the recent treatment advancements using the combination of a CDK4/6 inhibitor and an AI, approximately 50% of patients pass away within five years after an MBC diagnosis [7,8,9,10]. Therefore, it is essential to determine the optimal treatment sequence after the use of a CDK4/6 inhibitor in order to improve survival outcomes.

There are other new therapeutic strategies beyond the CDK4/6 inhibitor. For instance, the combined use of a PI3Kα-specific inhibitor (alpelisib) with ET has significantly longer PFS in *PIK3CA* mutant HR+HER2− MBC as the second-line treatment [13,14]. In addition, trastuzumab deruxtecan (T-Dxd) has a survival benefit in HR+HER2− low MBC included in HR+HER2− MBC [15]. However, genetic tests such as *PIK3CA* cannot be performed in real-time, and some of these novel therapies are not widely available.

In this study, we analyzed a consecutive retrospective cohort of patients with HR+/HER2− MBC who were treated first-line with palbociclib and letrozole. Our aim was to evaluate real-world long-term survival outcomes of palbociclib with letrozole as a first-line therapy, second-line treatment patterns after palbociclib, factors influencing treatment choice, and the prognosis associated with each selected treatment.

## 2. Materials and Methods

### 2.1. Patients

We collected patient data from the clinical data warehouse (CDW) in Samsung Medical Center (SMC). We selected data from HR+HER2− MBC patients who were treated with palbociclib and letrozole as the first-line treatment in metastatic setting therapy between January 2014 and December 2020. The diagnostic studies for MBC included chest computed tomography (CT), abdomino-pelvic CT, bone scan, or positron emission tomography-CT and brain imaging if indicated. HR and HER2 status examinations were permitted in metastatic biopsies as well as archival tissues. The details of BC pathology analysis were described in a previous study [16].

For analysis of second-line treatments, we excluded patients who had been lost to follow-up after first-line palbociclib with letrozole or the first cycle of second-line treatment.

### 2.2. Statistical Analysis

PFS was defined as the time from initiation of palbociclib with letrozole to disease progression or death from any cause, whichever occurred first. In addition, PFS2 was defined by the time from initiation of the second-line treatment to disease progression or death. The OS was defined as the time from initiation of palbociclib with letrozole to death from any cause, and the OS2 was the time between the start of the second-line treatment and death. PFS and OS were analyzed using the Kaplan–Meier method. Cox proportional hazard regression was used to estimate the hazard ratios and 95% confidence intervals (CIs). Correlations between clinical characteristics and tumor response were analyzed using a two-sided Student’s *t*-test and Fisher’s exact test. Two-tailed *p*-values < 0.05 were considered statistically significant. All statistical analyses were performed using IBM SPSS Statistics, ver. 29 (IBM Co., Armonk, NY, USA).

## 3. Results

### 3.1. Patient Baseline Characteristics and Updated Survival Analysis

We included 305 patients with a data cut-off date of 3 April 2023 (Figure 1). By that date, 181 patients (59%) experienced disease progression after first-line therapy with palbociclib and letrozole. Among these 181 patients, three died and five were lost to follow-up at that time of disease progression. Among the remaining 173 patients who received second-line treatment for MBC in SMC, seven were lost to follow-up after the first cycle of the second-line treatment. Ultimately, we included 166 patients in the second-line treatment analysis.

The baseline patient characteristics according to disease progression are described in Appendix A. In this analysis, age and ECOG performance status and de novo disease were not different between the two groups. However, visceral metastasis, number of metastatic sites, germline BRCA status, and initial CA-15-3 and CEA levels were different between the two groups; patients who had experienced disease progression for palbociclib with letrozole or not, respectively (*p* < 0.05, respectively).

In terms of survival analysis, 181 cases of disease progression and 62 deaths were observed during a median follow up of 41.7 months (interquartile range [IQR]: 33.6, 50.1). In this survival analysis, the median PFS was 29.0 months (95% confidence interval [CI]:23.5, 34.4). The median OS was not reached (Figure 2A,B). The five-year OS rate was 66.5% (Figure 2B).

### 3.2. Second-Line Treatment after Palbociclib with Letrozole

Capecitabine was the most frequently prescribed drug in our cohort (Appendix A). Of the 166 patients, 47 (28.5%) were treated with capecitabine after a CDK4/6 inhibitor, followed by exemestane with everolimus (27.3%) or cytotoxic chemotherapy other than capecitabine (18.8%) and fulvestrant-based treatment or endocrine monotherapy (12.7%). Eighteen patients (10.9%) were enrolled in clinical trials.

The reasons for each second-line treatment choice are described in Table 1. We evaluated the impact of clinical characteristics including disease progression site (Appendix A) and response duration of first-line palbociclib with letrozole. The response duration cut off was set at 12 months according to the definition of ER-driven disease in the era of CDK4/6 inhibitors [17]. Before the analysis, we excluded two patients who received intrathecal chemotherapy only. Of the remaining 164 patients, 64 developed disease progression in visceral organs (39.0%), 47 in the bone only (28.7%), and 53 at others (32.3%) (Table 1). In patients with visceral organ disease progression, capecitabine (31.3%) or cytotoxic chemotherapy (31.3%) was the most common second-line treatments followed by exemestane with everolimus (21.9%). Most of the 47 patients with bone metastasis alone were treated with exemestane with everolimus (38.2%), followed by capecitabine (23.4%) and fulvestrant (21.3%) (*p* = 0.008). The response duration of palbociclib with letrozole did not influence the choice of the second-line treatment strategy (*p* = 0.209). Other clinical factors including eastern cooperative oncology group (ECOG) performance status (*p* = 0.001), initial visceral metastasis (*p* = 0.006) and endocrine resistance in adjuvant setting (*p* = 0.011) were associated with second-line treatment strategies.

### 3.3. Progression-Free Survival and Overall Survival with Second-Line Treatment

We analyzed the survival outcomes of 164 patients. The PFS of second-line treatment (PFS2) was 4.87 months (IQR: 2.65, 9.45) (Figure 2C). The OS of second-line treatment (OS2) was 35.0 months. One hundred thirty-six patients experienced disease progression after second-line treatment (Figure 2D). With regard to treatment strategies, the median PFS2 was 7.4 months with capecitabine treatment, 5.2 months with exemestane and everolimus, 4.8 months with cytotoxic chemotherapy, 3.6 months with fulvestrant-based treatment, and 3.6 months with clinical trial enrollment (*p* = 0.066) (Figure 3A). The median OS2 was 42.3 months with capecitabine, 35.7 months with fulvestrant, 30.7 months with exemestane and everolimus, and 23.1 months with cytotoxic chemotherapy. The median OS2 for those in clinical trials was not reached (*p* = 0.064) (Figure 3B).

The clinical characteristics affecting PFS2 and OS2 were analyzed (Table 2 and Figure 3C–F). Favorable PFS2 was found in ER-driven tumors (hazard ratio: 0.73, 95% CIs: 0.51, 1.04; *p* = 0.078) and bone only progression (hazard ratio: 0.63, 95% CIs: 0.40, 1.00; *p* = 0.039). In contrast, poor PFS2 was associated with everolimus and exemestane treatment, or fulvestrant-based treatment (hazard ratio of everolimus with exemestane: 1.67, 95% CIs: 1.04, 2.67, hazard ratio of fulvestrant-based treatment: 2.38, 95% CIs: 1.36, 4.16; *p* = 0.031). In terms of OS2, PFS2 was the strongest prognostic factor of OS2 (hazard ratio: 0.32, 95% CIs: 0.19, 0.59; *p* < 0.001). In addition, ER-driven BC was associated with good OS2 (hazard ratio: 0.53, 95% CIs: 0.31, 0.90; *p* = 0.019), whereas initial visceral metastasis and visceral organ disease progression were associated with poor OS2 (hazard ratio: 2.10, 95% CIs: 1.04, 4.23; *p* = 0.039 and hazard ratio: 2.34, 95% CIs: 1.17, 4.23; *p* = 0.026).

### 3.4. Effect of Second-Line Treatment According to Site of Disease Progression

The second-line treatment regimens were decided based on the organ of disease progression after first-line therapy. Therefore, we evaluated the effectiveness of second-line treatment according to the sites of disease progression (Figure 4).

In patients with visceral organ disease progression, capecitabine and cytotoxic chemotherapy (except capecitabine) had 5.53 and 5.17 months in PFS2, respectively, while fulvestrant had 0.90 months. However, the second-line treatment strategies were not associated with PFS2 (*p* = 0.640) (Figure 4A) or OS2 (*p* = 0.583) (Figure 4B). However, patients treated with fulvestrant or other endocrine alone treatments only had 11.5 months of OS2 compared to 32.3 months with capecitabine and 28.2 months with exemestane and everolimus.

In the 47 patients with bone metastasis alone, capecitabine had superior PFS2 than did other second-line regimens (*p* = 0.011) (Figure 4C). The median PFS2 was 15.4 months with capecitabine, 8.33 months with exemestane and everolimus, 6.63 months with endocrine only treatment, and 3.67 months with clinical trial enrollment. The two patients who were treated with cytotoxic chemotherapy (excluding capecitabine) had only 1.4 months of PFS2. The OS2 was not reached in this population. However, OS2 differed according to second-line treatment strategy (*p* = 0.002) (Figure 4D).

Fifty-three patients had disease progression in locations other than bone and visceral organs; the second-line regimens in these patients marginally affected their PFS2, although without statistical significance (*p* = 0.096) (Figure 4E). In this population, capecitabine also had the longest PFS2 (5.70 months) compared to that of the other regimens. The OS2 did not differ according to the second-line treatment (*p* = 0.826) (Figure 4F).

### 3.5. Clinical Characteristics Affecting Overall Survival

The clinical factors that affected the OS were analyzed in all 305 patients (Table 3). In this analysis, visceral metastasis (hazard ratio: 1.60, 95% CIs: 0.97, 2.65; *p* = 0.068), initial CA-15-3 elevation (hazard ratio: 1.92, 95% CIs: 1.18, 3.12), endocrine resistance (hazard ratio for primary resistance: 2.25, 95% CIs: 1.17, 4.32), number of metastatic organs (hazard ratio: 1.774, 95% CIs: 1.091, 2.886), and germline BRCA mutation (hazard ratio: 3.99, 95% CIs: 1.31, 12.14) were all associated with a short duration of OS (*p* < 0.05, respectively). We also evaluated the association between clinical characteristics and OS in 164 patients who had disease progression after first line treatment with a CDK4/6 inhibitor and letrozole (Table 4). In that analysis, we found that OS was influenced by initial visceral metastasis (hazard ratio:2.20; *p* = 0.027), number of initial metastatic organs (hazard ratio:1.71, *p* = 0.048), ER driven tumor (hazard ratio:0.19, *p* < 0.001), and visceral organ disease progression after a CDK4/6 inhibitor (hazard ratio: 2.64, *p* = 0.020). The initial CA-15-3 status, germline BRCA status, and second-line treatment strategies did not affect the OS.

## 4. Discussion

This up-to-date survival analysis found that first-line treatment of HR+HER2− MBC patients with palbociclib and letrozole provided a median PFS of 29.0 months. The median OS was not reached by the median follow up of 41.7 months. In addition, the five-year OS rate was 66.5%.

PALOMA-2, the pivotal clinical trial for the first-line CDK4/6 inhibitor with AI in HR+HER2− MBC patients, reported a median PFS of 24.8 months [3]. The Asian subgroup analysis in PALOMA-2 found a consistent effect of palbociclib and a median PFS of 25.7 months [18]. In real world data, the median PFS of palbociclib was 19.8 months [11]. In our prior study, we found a median PFS of 28.7 months, which was consistent with our current results [16]. However, the OS benefit of palbociclib has not been consistent among previous studies. In PALOMA-2 clinical trial, the OS was 53.9 months in those treated with palbociclib plus an AI compared to 51.2 months in those treated with an AI alone (*p* = 0.338). In contrast, real world data found an OS of 57.8 months in those treated with palbociclib plus an AI compared to 43.5 months in those treated with an AI alone (*p* < 0.001) [7,11]. In our data, the median OS was not reached, and the five-year OS rate was 66.5%.

Before the era of CDK4/6 inhibitors, non-steroidal AI was considered the first-line treatment for HR+HER2− MBC without visceral crisis [19]. Previous clinical trials, as well as real world data analyses for HR+HER2− MBC, found that AI as a first-line therapy provided 3–4 years of OS [20,21,22,23,24]. Consistently, real world data with palbociclib showed 43.5 months of OS in patients treated with AI as the first-line treatment [11].

Recent treatment advances have forced OS improvement in HR+HER2− MBC. In particular, implementation of CDK4/6 inhibitors has significantly changed PFS and OS in this population compared to those in the era before these medications. A recent cohort study suggested that the median OS of HR+HER2− MBC from 2017–2019 was 38.4 months, and that the death risk decreased by 24% compared to that before 2017 [23]. In addition, this cohort study found that the OS was influenced by the age at MBC diagnosis, metastatic location, and number of metastatic sites. Consistent with prior findings, we found that the OS was affected by visceral metastasis and number of metastatic sites. The initial elevation of the serum CA-15-3 and germline BRCA alteration increased the risk of death. Moreover, we attempted to evaluate the association between tumor response to treatment and OS. In this analysis, initial visceral metastasis and number of metastases still affected the OS, whereas the ER driven BC and disease progression site after treatment with palbociclib and letrozole were added as a factor that affects the OS. However, the effect of germline BRCA status has disappeared. These results suggest an interaction between tumor biology and treatment that dynamically affects patient prognosis, while some clinical characteristics of MBC still impact the OS.

In our cohort, capecitabine was the most commonly used drug for second-line treatment. Although capecitabine is considered cytotoxic chemotherapy, it is an oral anti-metabolite that is quite tolerable and active in MBC patients [25,26]. Recent clinical trials with palbociclib and ET used capecitabine as the direct competitor in HR+HER2− MBC patients, and capecitabine achieved good survival outcomes [27,28]. In our study, we preferred capecitabine as the second-line treatment in visceral disease progression or other progression (not including bone-only progression). In addition, capecitabine has superior PFS2 compared to those of other agents in multivariate analysis. Indeed, capecitabine also had longer PFS2 duration rather than that of other agents in bone-only progression.

In bone-only progression, fulvestrant was preferred as the second-line treatment. Fulvestrant is a selective estrogen receptor down regulator (SERD) that inhibits ESR1 activity and proteolytic stability. Therefore, mutant *ESR1*, which is the most common resistance mechanism for AI treatment in HR+HER2− BC, could be effectively inhibited by fulvestrant [29,30]. However, fulvestrant was rarely effective in HR+HER2− MBC after treatment with a CDK4/6 inhibitor and AI in previous clinical trials. Fulvestrant produced a PFS of 1.98 months in the VERONICA clinical trial and of 3.6 months in the CAPItello-291 trial [31,32]. Consequentially, fulvestrant led to 3.67 months of PFS in this study; however, we observed 6.63 months of PFS in fulvestrant and endocrine treatment only in bone lesion progression. Therefore, second-line treatment should be selected with regard to the sites of BC progression.

ER-driven tumors affected the OS and the OS2 but not the PFS2. ER-driven tumors tend to have a relatively long response duration for endocrine therapy and remain sensitive to ET. Therefore, we suggest that ER-driven tumors have a favorable survival outcome regardless of the PFS2 and second-line treatment strategies. In addition, PFS2 also affected to the OS and the OS2. This meant that PFS2 was also significant prognostic factor regardless of ER-driven tumor or not. Moreover, visceral organ disease progression was also significantly associated with OS and the OS2 as well as initial visceral metastasis. This suggested that the dynamic interaction between tumor itself and treatment strategy would impact to patients’ survival. Eventually, second line treatment strategy would be important to treat HR+HER2− MBC patients as well as the first line treatment strategy.

In this real-world study, we rarely performed tissue biopsy after disease progression of CDK4/6 inhibitor. As BC subtype change occurs in 20% of MBC, and the most common mechanism of subtype change is loss of ER and PgR, BC biopsy after disease progression is important to guide second-line treatment strategies. Furthermore, genetic testing can be used to precisely treat patients with progressive MBC. However, tissue biopsy has been rarely performed in real clinic, and therefore, the lack of biopsies and genetic testing was a limitation in our data analysis. We expect that further genome and real time subtype-based treatments would improve survival outcomes in HR+HER2− MBC patients.

Recent advancements in BC treatment have been achieved using genetic information. Targetable genetic alterations are found in ~30–40% and ~5% of HR+/HER2− BC [33,34]. *PIK3CA* is the most commonly altered gene and is also targetable. *RB1* loss and the *ESR1* mutation are associated with resistance to treatment with a CDK4/6 inhibitor and AI [35,36]. Future treatment strategies will include genetic information and targeted agents, including PIK3a inhibitors and next-generation SERDs [13,14,37].

In this study, we chose the second-line treatment based on initial visceral metastasis and the sites of disease progression. Second line treatment affected to the PFS2 but not the OS and the OS2. However, PFS2 was significantly associated with the OS and OS2, and furthermore, our cohort had good OS outcomes compared to those of other previous studies using Palbociclib [7,11]. Therefore, our findings suggest that each line of treatment is important to improve the OS in HR+HER2− MBC.

## 5. Conclusions

In conclusion, optimizing the sequence of later-line treatments based on MBC status is important to improve prognosis in patients with HR+/HER2− MBC, especially given the recent standardization of first-line treatment and the many available second-line options.

## Figures and Tables

**Figure 1 cancers-15-03431-f001:**
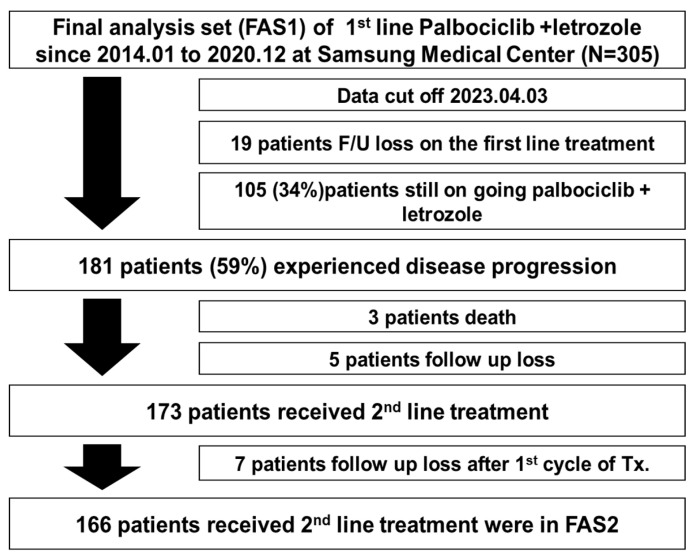
Consort diagram.

**Figure 2 cancers-15-03431-f002:**
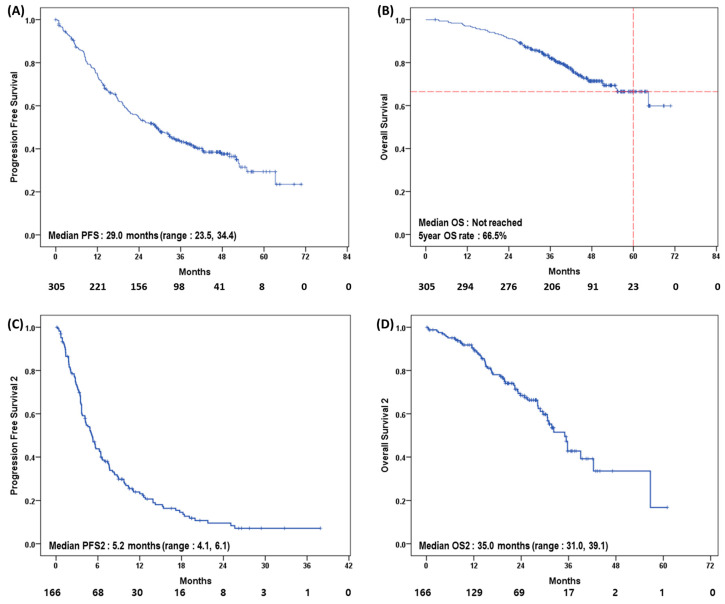
(**A**) Median progression free survival (PFS), (**B**) median overall survival (OS), (**C**) median PFS2, (**D**) median OS2 in all patients (N = 305).

**Figure 3 cancers-15-03431-f003:**
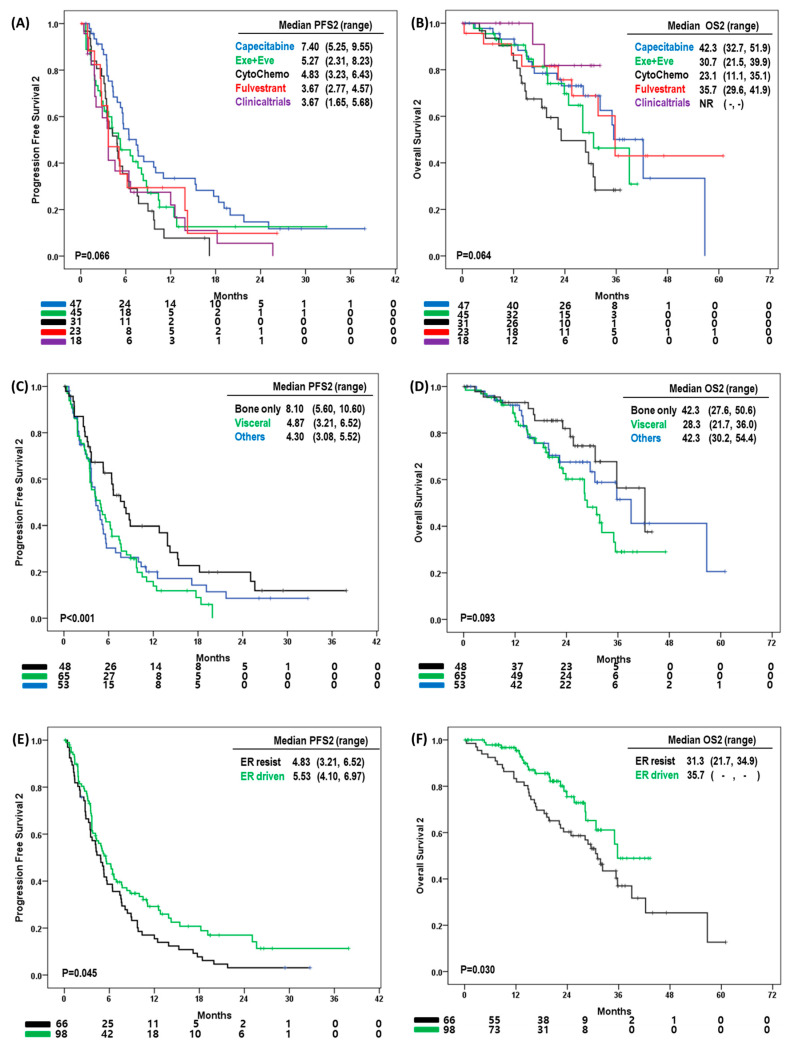
(**A**) Median progression free survival 2 (PFS2), (**B**) median overall survival (OS2) according to second line treatment strategies, (**C**) median PFS2 (**D**) median OS2 according to the sites of disease progression, (**E**) median PFS2, and (**F**) median OS2 according to ER driven tumor.

**Figure 4 cancers-15-03431-f004:**
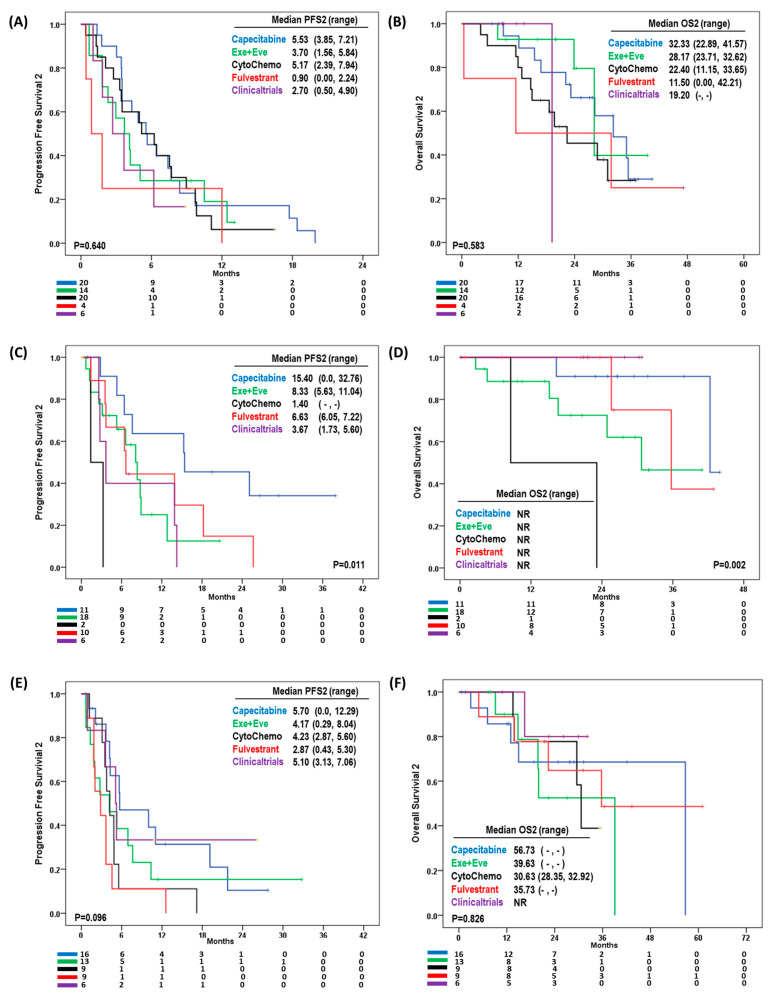
(**A**) Median progression free survival 2 (PFS2), (**B**) median overall survival 2 (OS2) in patients with visceral organ disease progression according to treatment strategies, (**C**) median PFS2, (**D**) median OS2 in patients with bone only disease progression according to treatment strategies, (**E**) median PFS2 and (**F**) median OS2 in patients with other sites’ disease progression according to treatment strategies.

**Table 1 cancers-15-03431-t001:** Relationships between clinical factors and the second-line treatment regimen (N = 164).

	Capecitabine	Eve/Exe	Cytotoxic	Fulvestrant	Clinical Trial	*p*-Value
Age						0.292
<50YO ^1^ (n = 74)	19	26	12	8	9	
>50YO (n = 90)	28	19	19	15	19	
ECOG PS ^2^						0.001
0 (n = 98)	30	32	23	9	4	
1–2 (n = 66)	17	13	8	14	14	
Initial ER ^3^ score						0.971
Strong (n = 147)	42	40	29	20	16	
Weak (n = 17)	5	5	2	3	2	
Initial PgR ^4^ score						0.477
Strong (n = 64)	17	13	17	6	11	
Weak (n = 63)	19	19	10	11	4	
No (n = 37)	11	13	4	6	3	
Initial Ki-67 score						0.326
1+ (n = 103)	31	28	15	16	13	
2+ (n = 46)	12	14	13	4	3	
3+ (n = 12)	3	2	2	3	2	
4+ (n = 3)	1	1	1	0	0	
Initial visceral metastasis						0.006
No (n = 120)	26	39	21	20	14	
Yes (n = 44)	21	6	10	3	4	
Number of metastatic sites						0.262
1 (n = 74)	16	20	13	15	10	
2 (n = 61)	18	20	12	5	6	
3 or more (n = 29)	13	5	6	3	2	
Endocrine resistance in adjuvant setting						0.011
De novo (n = 70)	22	14	11	12	11	
Primary resistance ^5^ (n = 28)	14	4	4	5	1	
Secondary resistance ^6^ (n = 27)	4	12	6	1	4	
No ET resistance (n = 39)	7	15	10	5	2	
Disease progression sites						0.008
Others (n = 53)	16	13	9	9	6	
Visceral meta (n = 64)	20	14	20	4	6	
Bone only (n = 47)	11	18	2	10	6	
PFS ^7^ of the first line treatment						0.209
ER driven (n = 98)	25	28	15	16	14	
Not driven (n = 66)	22	17	16	7	4	

^1^: years of age; ^2^: performance status; ^3^: Estrogen receptor; ^4^: Progesterone receptor; ^5^: Disease recurrence before 24 months of adjuvant endocrine therapy; ^6^: disease recurrence between after 24 months of adjuvant endocrine therapy and after 12 months of the end of endocrine therapy; ^7^: progression free survival.

**Table 2 cancers-15-03431-t002:** Clinical characteristics that affected progression-free survival 2 and overall survival 2 (N = 164).

Factors for PFS2	Ref	N	Hazard Ratio	95% CI		*p*-Value
ECOG PS ^1^	0	98				0.793
1–2		66	1.054	0.710	1.566	
Endocrine resistance	De novo	70				0.778
Primary resistance		28	1.000	0.603	1.660	
Secondary resistance		27	0.814	0.488	1.358	
No resistance		39	0.817	0.511	1.307	
Initial visceral metastasis	No	120				0.596
Yes		44	1.166	0.717	1.896	
ER ^2^-driven BC ^3^		66				0.078
Yes	No	98	0.725	0.507	1.036	
Disease progression site		53				0.039
Visceral organ	Other	64	1.169	0.773	1.768	
Bone only		47	0.633	0.397	1.000	
Second-line treatment	Capecitabine	47				0.031
Everolimus/exemestane		45	1.665	1.038	2.670	
Other cytotoxic chemo		31	1.655	0.995	2.753	
Fulvestrant		23	2.383	1.364	4.163	
Clinical trials		18	1.713	0.904	3.246	
**Factors for OS2**	**Ref**	**N**	**Hazard Ratio**	**95% CI**		***p*-Value**
ECOG PS ^1^	0	98				0.720
1–2		66	0.897	0.494	1.628	
Endocrine resistance	De novo	70				0.759
Primary resistance		28	0.770	0.373	1.593	
Secondary resistance		27	0.687	0.297	1.590	
No resistance		39	0.980	0.465	2.065	
Initial visceral metastasis	No	120				0.039
Yes		44	2.097	1.039	4.234	
ER ^2^-driven BC ^3^	No	66				0.019
Yes		98	0.525	0.306	0.901	
Disease progression site	Other	53				0.026
Visceral organ		64	2.339	1.166	4.234	
Bone only		47	0.967	0.461	2.027	
Second-line treatment	Capecitabine	47				0.316
Everolimus/exemestane		45	0.890	0.421	1.878	
Other cytotoxic chemo		31	1.253	0.600	2.617	
Fulvestrant		23	0.669	0.274	1.631	
Clinical trials		18	0.292	0.064	1.328	
PFS2 ^4^	≤5.2 months	88				<0.001
>5.2 months		76	0.323	0.188	0.589	

^1^: Eastern Cooperative Oncology Group Performance Status; ^2^: Estrogen receptor; ^3^: Breast cancer; ^4^: progression free survival 2.

**Table 3 cancers-15-03431-t003:** Clinical characteristics that affected overall survival (N = 305).

Factors	Ref	N	Hazard Ratio	95% CI		*p*-Value
Age	<50	130				0.131
>50 years old		175	1.493	0.888	2.510	
ECOG PS ^1^	0	177				0.658
1		122	0.793	0.483	1.301	
2		5	1.543	0.434	5.484	
Unknown		1	-	-	-	
Visceral metastasis	No	239				0.068
Yes		66	1.599	0.967	2.645	
Initial CA-15-3	Normal	272				0.019
Elevation		31	1.922	1.184	3.120	
Unknown		2	0.706	0.67	2.986	
Initial CEA	Normal	272				0.908
Elevation		31	0.987	0.550	1.771	
Unknown		2	1.505	0.231	9.790	
Endocrine resistance	De novo	70				0.031
Primary resistance		28	2.250	1.171	4.323	
Secondary resistance		27	0.854	0.427	1.705	
No resistance		39	0.898	0.499	1.578	
Germline BRCA status	Normal	92				0.040
Mutation		6	3.989	1.311	12.139	
Not tested		207	1.460	0.867	2.459	
Number of meta organs	1	157				0.021
2 or more		148	1.774	1.091	2.886	

^1^: Eastern Cooperative Oncology Group Performance Status.

**Table 4 cancers-15-03431-t004:** Clinical characteristics that affected the OS in patients receiving second-line treatment (N = 164).

Factors for OS	Ref	N	Hazard Ratio	95% CI		*p*-Value
Visceral metastasis	No	120				0.062
Yes		44	1.935	0.968	3.867	
Initial CA-15-3	Normal	81				0.609
Elevation		74	1.295	0.762	2.200	
Unknown		9	0.924	0.208	4.095	
Germline BRCA status	Wild type	54				0.604
Mutation		6	1.571	0.475	5.199	
Not tested		104	0.899	0.482	1.674	
Number of meta organs	1	74				0.048
2 or more		90	1.705	1.004	2.894	
ER-driven BC	No	66				<0.001
Yes		98	0.180	0.103	0.316	
Disease progression site	Other	53				0.034
Visceral organ		64	2.191	1.099	4.368	
Bone only		47	0.932	0.444	1.956	
Second-line treatment	Capecitabine	47				0.373
Everolimus/exemestane		45	0.984	0.469	2.063	
Other cytotoxic chemo		31	1.318	0.624	2.786	
Fulvestrant		23	0.829	0.353	1.947	
Clinical trials		18	0.292	0.065	1.318	
PFS2 ^1^	≤5.2 months	88				<0.001
>5.2 months		76	0.340	0.194	0.598	

^1^: Progression Free Survival 2.

## Data Availability

A data sharing statement provided by the authors is available by request to the corresponding author.

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
