# Peer review of "Real World Experience of Second-Line Treatment Strategies after Palbociclib and Letrozole: Overall Survival in Metastatic Hormone Receptor-Positive Human Epidermal Growth Factor Receptor 2-Negative Breast Cancer"

_cancers, 2023, doi:10.3390/cancers15133431_

Round 1
Reviewer 1 Report
Review of:
Real world experience of second-line treatment strategies after palbociclib and letrozole: Overall survival in metastatic hormone receptor-positive human epidermal growth factor receptor 2-negative breast cancer
Comments to the Authors
The authors present real world data for the considerations for second line treatments after CDk4/6 aromatase inhibitor treatment.
They conclude that for the first-line treatment most significant factors for OS are visceral mets, elevated CA-15-3, BRCA status and number of metastatic organs.
Second-line treatments OS factors were visceral mets, number of met. Organs, ER-driven, visceral vs bone, treatments.
Capecitabine showed the best OS and PFS even in bone met patients, although Fulvestrant was preferred.
In summary, the authors showed a comprehensive analysis of their patient data.
Detailed comments/questions:
What is the treatment in the cohort “clinical trials”?
Where any bone mets observed in the first-line treatment? If yes please include them in supplemental table 1
What are the “other” met sites, besides visceral and bone. Any information?
In Table 4: The number of mets are a significant as well as “visceral” vs “only bone”. Can those factors be stratified in more detailed fashion?
Line 148-149: This sentence does not look complete. “… was influenced (…).” [by what?]
Minor editing required
Author Response
Reviewer 1.
The authors present real world data for the considerations for second line treatments after CDk4/6 aromatase inhibitor treatment. They conclude that for the first-line treatment most significant factors for OS are visceral mets, elevated CA-15-3, BRCA status and number of metastatic organs. Second-line treatments OS factors were visceral mets, number of met. Organs, ER-driven, visceral vs bone, treatments. Capecitabine showed the best OS and PFS even in bone met patients, although Fulvestrant was preferred. In summary, the authors showed a comprehensive analysis of their patient data.
Detailed comments/questions:
What is the treatment in the cohort “clinical trials”?
: Thanks for your comments. In our cohort, 18 patients were enrolled in clinical trial as the second line treatment. Of 18 patients, 10 were enrolled in CAPitello-291, two in PETRA, two in CO23867, two in EMBER, one in C0541001 and one in ACT16105.
The information of each clinical trial is described as below.
|
Trial Name |
NCT nurmber |
Title |
|
CAPitello-291 |
NCT04305496 |
Capivasertib+Fulvestrant vs Placebo+Fulvestrant as Treatment for Locally Advanced (Inoperable) or Metastatic HR+/HER2- Breast Cancer |
|
PETRA |
NCT04644068 |
Study of AZD5305 as Monotherapy and in Combination With Anti-cancer Agents in Patients With Advanced Solid Malignancies (PETRA) |
|
CO42867 |
NCT04802759 |
A Study Evaluating the Efficacy and Safety of Multiple Treatment Combinations in Participants With Breast Cancer |
|
EMBER |
NCT04188548 |
A Study of LY3484356 in Participants With Advanced or Metastatic Breast Cancer or Endometrial Cancer |
|
C0541001 |
NCT03284723 |
PF-06804103 Dose Escalation in HER2 Positive and Negative (Negative Only in Part 2) Solid Tumors |
|
ACT16105 |
NCT04059484 |
Phase 2 Study of Amcenestrant (SAR439859) Versus Physician's Choice in Locally Advanced or Metastatic ER-positive Breast Cancer |
Where any bone mets observed in the first-line treatment? If yes please include them in supplemental table 1.
: We already presented metastatic lesions at diagnosis in our previous study (DOI: 10.3389/fonc.2021.759150). According to disease progression after palbociclib with letrozole, 57 patients who had initially bone metastasis only have underwent disease progression whereas 46 patients did not undergo disease progression. We added this information in supplementary table 1.
Supplementary Table 1. Baseline patient characteristics (N=305)
|
Metastatic sites |
|
|
<.001 |
|
Visceral |
52 (28.7) |
14 (11.3) |
|
|
Liver |
48 (26.5) |
12(9.7) |
|
|
Non-visceral |
129 (71.3) |
110 (88.7) |
|
|
Bone only |
57 (31.5) |
46 (37.1) |
|
What are the “other” met sites, besides visceral and bone. Any information?
: Other metastatic sites are vary and therefore we categorized all metastatic lesions into three; visceral, bone only and others. According to your comment, we added the information of disease progression sites in supplementary table 3.
Supplementary Table 3. Disease progression sites after palbociclib with letrozole (N=164)
|
Organ |
N (%) |
|
Visceral |
64 (39.0%) |
|
Liver |
61 (37.2%) |
|
Central nerve system |
4 (2.4%) |
|
Lymphangitic lung metastases |
1 (0.6%) |
|
Non-visceral metastasis |
53 (32.3%) |
|
Lung |
11 (6.7%) |
|
Lymph nodes |
27 (16.5%) |
|
Breast |
12 (7.3%) |
|
Pleura |
8 (4.9%) |
|
Ovary |
1 (0.6%) |
|
Stomach |
1 (0.6%) |
|
Bone only |
47 (28.7%) |
In Table 4: The number of mets are a significant as well as “visceral” vs “only bone”. Can those factors be stratified in more detailed fashion?
: We also analyzed our data according to your comment. Before analysis, we check the number of metastatic sites.
|
Number of sites |
N(%) |
|
1 |
74 (45.1%) |
|
2 |
62 (37.8%) |
|
3 |
18 (11.0%) |
|
4 |
5 (3.0%) |
|
5 |
3 (1.8%) |
|
7 |
2 (1.2%) |
Therefore, we divided the number of sites into 3 categories : one, two, three or metastatic sites. The result of analysis was described as below.
|
Factors for OS |
Ref |
N |
Hazard ratio |
95% CI |
|
P-value |
|
Visceral metastasis |
No |
120 |
|
|
|
0.016 |
|
Yes |
|
44 |
2.415 |
1.181 |
4.926 |
|
|
Initial CA-15-3 |
Normal |
81 |
|
|
|
0.239 |
|
Elevation |
|
74 |
1.420 |
0.831 |
2.425 |
|
|
Unknown |
|
9 |
0.525 |
0.114 |
2.424 |
|
|
Germline BRCA status |
Wild type |
54 |
|
|
|
0.975 |
|
Mutation |
|
6 |
1.065 |
0.295 |
3.848 |
|
|
Not tested |
|
104 |
0.948 |
0.493 |
1.926 |
|
|
Number of meta organs |
1 |
74 |
|
|
|
0.045 |
|
2 |
|
62 |
2.101 |
1.148 |
3.844 |
|
|
3 or more |
|
28 |
1.762 |
0.867 |
3.584 |
|
|
ER-driven BC |
No |
66 |
|
|
|
<0.001 |
|
Yes |
|
98 |
0.155 |
0.085 |
0.284 |
|
|
Disease progression site |
Other |
53 |
|
|
|
0.022 |
|
Visceral organ |
|
64 |
1.952 |
0.984 |
3.875 |
|
|
Bone only |
|
47 |
0.712 |
0.343 |
1.480 |
|
|
Second-line treatment |
Capecitabine |
47 |
|
|
|
0.253 |
|
Everolimus/exemestane |
|
45 |
1.255 |
0.605 |
2.605 |
|
|
Other cytotoxic chemo |
|
31 |
1.910 |
0.953 |
3.828 |
|
|
Fulvestrant |
|
23 |
2.014 |
0.821 |
4.941 |
|
|
Clinical trials |
|
18 |
0.514 |
0.116 |
2.273 |
|
This table has similar results of previous Table 4. In addition, number of metastatic sites two and three have similar hazard ratio compared to only one metastatic site.
Line 148-149: This sentence does not look complete. “… was influenced (…).” [by what?]
: Thanks for your comment. We revised our manuscript according to your comment.
Lines 152-153 : “The response duration of palbociclib with letrozole did not influence the choice of the second-line treatment strategy (p=0.209).”

Reviewer 2 Report
This manuscript presents an interesting study regarding the second-line treatment strategies after use of palbociclib and letrozole in hormone receptor-positive (HR+), HER2-negative (HER2−) metastatic breast cancer (MBC). The authors used real-world clinical data, and analyzed the second-line treatment regimens and progression-free and overall survival. The study is important. However, there are several points that might be addressed to clarify the findings and improve the manuscript. Major comments: 1. What percentage of patients with HR+HER2− MBC were treated with palbociclib and letrozole as the first-line treatment? 2. The term “Not progression” would be better changed to “No progression” throughout. 3. Please clearly note the definition of the terms “Progression” and “No progression” in Supplementary Table 1. For how many years were patients with “No progression” followed up? Moreover, adding information on the contents of adjuvant treatments and the median follow-up time is recommended as characteristics in this Table. 4. When considering the treatment strategies and the progression-free and overall survival in the second-line treatment, it may also be important to consider age, ECOG performance status, expression levels of ER/PgR and Ki-67, contents of adjuvant treatments, and disease-free interval, in addition to the information on factors such as metastatic sites and the duration of the first-line therapy. Please provide these factors as clinical factors in Tables 1 to 4, regardless of the influence of these factors. 5. In the part about “Factors for OS2” in Table 2 and in Table 4, please provide PFS2 as a factor. 6. In Table 3, please provide the characteristics that were selected in Supplementary Table 1, information on the duration of the first-line therapy (or ER-driven tumors), and PFS2 as factors. 7. Please show a summary of the second-line treatment strategy, including the factors that the authors recommend should be considered after palbociclib and letrozole, in a figure in the Discussion section. Minor comments: 1. Line 28 and line 33 in the Abstract; The full names for “PFS2” and “OS2” should be provided. 2. Line 52; “And” should be deleted. 3. Line 289; “…. in patients treated with AI alone as the first-line treatment.” Is the term “alone” necessary?Author Response
Reviewer 2.
This manuscript presents an interesting study regarding the second-line treatment strategies after use of palbociclib and letrozole in hormone receptor-positive (HR+), HER2-negative (HER2−) metastatic breast cancer (MBC). The authors used real-world clinical data, and analyzed the second-line treatment regimens and progression-free and overall survival. The study is important. However, there are several points that might be addressed to clarify the findings and improve the manuscript.
Major comments:
- What percentage of patients with HR+HER2− MBC were treated with palbociclib and letrozole as the first-line treatment?
: In Korea, CDK4/6 inhibitor with aromatase inhibitor can be used as the first line treatment only in metastatic setting and regulated by Ministry of Food and Drug Safety. From July 2016, to December 2021, 465 patients received the first line endocrine treatment of HR+HER2- metastatic breast cancer in Samsung Medical Center. Of 465 patients, 305 patients were treated with palbociclib with letrozole followed by 65 patients with aromatase inhibitor alone, 63 patients with tamoxifen, 15 patients with palbociclib with fulvestrant, nine with abemaciclib with letrozole, five with ribociclib with letrozole and three with GnRH agonist single. This situation depended on the regulation of Ministry of Food and Drug Safety.

- The term “Not progression” would be better changed to “No progression” throughout.
: Thanks for your consideration. We revised our manuscript according to your comment.
Supplementary Table 1. Baseline patient characteristics (N=305)
|
Characteristic |
Progression (N=181) |
No progression (N=124) |
P-value |
- Please clearly note the definition of the terms “Progression” and “No progression” in Supplementary Table 1. For how many years were patients with “No progression” followed up? Moreover, adding information on the contents of adjuvant treatments and the median follow-up time is recommended as characteristics in this Table.
: Median follow up duration is 41.7 months in all patients; 44.7 months in patients who have not experienced disease progression and 39.8 months in patients. Of 305 patients, 39 patients had been received neoadjuvant chemotherapy and 164 patients had adjuvant chemotherapy.
According to your comment, we revised our manuscript.
Supplementary Table 1. Baseline patient characteristics (N=305)
|
Characteristic |
Progression (N=181) |
No progression (N=124) |
P-value |
|
Median FU1 (IQR)2 |
39.8 (30.7,49.7) |
44.7 (36.1, 50.1) |
|
|
Age |
|
|
.308 |
|
Median (range) |
51.5 (31.5, 78.1) |
52.0 (33.3, 86.7) |
|
|
<50 years old |
80 (44.2) |
50 (40.3) |
|
|
>50 years old |
101 (55.8) |
74 (59.7) |
|
|
ECOG PS3 |
|
|
.465 |
|
0 |
104 (57.5) |
73 (58.9) |
|
|
1 |
75 (41.4) |
47 (37.9) |
|
|
≥2 |
2 (1.1) |
3 (2.4) |
|
|
Unknown |
0 |
1 (0.8) |
|
|
Disease status |
|
|
.716 |
|
De novo |
62 (34.3) |
45 (36.3) |
|
|
Recurred |
119 (65.7) |
79 (63.7) |
|
|
Adjuvant CTx4 |
92 (50.8) |
71 (57.3) |
|
|
Neoadjuvant CTx |
27 (14.9) |
8 (6.4) |
|
|
Disease-free interval5 |
(n=198) |
|
.056 |
|
<12 months |
76 (63.9) |
39 (49.4) |
|
|
≥12 months |
43 (36.1) |
40 (50.6) |
|
|
Metastatic sites |
|
|
<.001 |
|
Visceral |
52 (28.7) |
14 (11.3) |
|
|
Liver |
48 (26.5) |
12(9.7) |
|
|
Non-visceral |
129 (71.3) |
110 (88.7) |
|
|
Bone only |
57 (31.5) |
46 (37.1) |
|
|
No. of disease sites |
|
|
.012 |
|
1 |
81 (44.8) |
77 (62.1) |
|
|
2 |
68 (37.6) |
32 (25.8) |
|
|
3 or more |
32 (17.6) |
15 (12.1) |
|
|
Germline BRCA status |
|
|
.041 |
|
Not tested |
115 (63.5) |
91 (73.4) |
|
|
BRCA1/2 mutation |
6 (3.3) |
0 |
|
|
No BRCA mutation |
60 (33.2) |
33 (26.6) |
|
|
Initial CA-15-3 status |
|
|
.001 |
|
Normal range |
91 (50.3) |
85 (68.5) |
|
|
Elevated |
81 (44.8) |
30 (24.2) |
|
|
Unknown |
9 (4.9) |
9 (7.3) |
|
|
Initial CEA status |
|
|
<0.001 |
|
Normal range |
128 (70.7) |
103 (83.1) |
|
|
Elevated |
49 (27.1) |
13 (10.5) |
|
|
Unknown |
4 (2.2) |
8 (6.4) |
|
1: follow up; 2: Interquartile range; 3: performance status; 4: chemotherapy; 5:defined as breast cancer recurrence within 1 year following adjuvant ET completion
- When considering the treatment strategies and the progression-free and overall survival in the second-line treatment, it may also be important to consider age, ECOG performance status, expression levels of ER/PgR and Ki-67, contents of adjuvant treatments, and disease-free interval, in addition to the information on factors such as metastatic sites and the duration of the first-line therapy. Please provide these factors as clinical factors in Tables 1 to 4, regardless of the influence of these factors.
: We performed further analysis according to your comment. In this analysis, ECOG performance status, initial visceral metastasis and primary endocrine resistance were affected to choice of 2nd line treatment after palbociclib with letrozole.
According to this analysis, we updated Table 1 and result section.
Table 1. Relationships between clinical factors and the second-line treatment regimen (N=164)
|
|
Capecitabine |
Eve/Exe |
Cytotoxic |
Fulvestrant |
Clinical trial |
P-value |
|
Age |
|
|
|
|
|
0.292 |
|
<50YO (n=74) |
19 |
26 |
12 |
8 |
9 |
|
|
>50YO(n=90) |
28 |
19 |
19 |
15 |
19 |
|
|
ECOG |
|
|
|
|
|
0.001 |
|
0 (n=98) |
30 |
32 |
23 |
9 |
4 |
|
|
1-2 (n=66) |
17 |
13 |
8 |
14 |
14 |
|
|
Initial ER score |
|
|
|
|
|
0.971 |
|
Strong (n=147) |
42 |
40 |
29 |
20 |
16 |
|
|
Weak (n=17) |
5 |
5 |
2 |
3 |
2 |
|
|
Initial PgR score |
|
|
|
|
|
0.477 |
|
Strong (n=64) |
17 |
13 |
17 |
6 |
11 |
|
|
Weak (n=63) |
19 |
19 |
10 |
11 |
4 |
|
|
No (n=37) |
11 |
13 |
4 |
6 |
3 |
|
|
Initial Ki-67 score |
|
|
|
|
|
0.326 |
|
1+ (n=103) |
31 |
28 |
15 |
16 |
13 |
|
|
2+ (n=46) |
12 |
14 |
13 |
4 |
3 |
|
|
3+ (n=12) |
3 |
2 |
2 |
3 |
2 |
|
|
4+ (n=3) |
1 |
1 |
1 |
0 |
0 |
|
|
Initial visceral metastasis |
|
|
|
|
0.006 |
|
|
No (n=120) |
26 |
39 |
21 |
20 |
14 |
|
|
Yes (n=44) |
21 |
6 |
10 |
3 |
4 |
|
|
Number of metastatic sites |
|
|
|
|
0.262 |
|
|
1 (n=74) |
16 |
20 |
13 |
15 |
10 |
|
|
2 (n=88) |
18 |
20 |
12 |
5 |
6 |
|
|
3 or more (n=29) |
13 |
5 |
6 |
3 |
2 |
|
|
Endocrine resistance in adjuvant setting |
|
|
|
|
|
0.011 |
|
De novo (n=70) |
22 |
14 |
11 |
12 |
11 |
|
|
Primary ET resistance (n=28) |
14 |
4 |
4 |
5 |
1 |
|
|
Secondary ET resistance (n=27) |
4 |
12 |
6 |
1 |
4 |
|
|
No ET resistance (n=39) |
7 |
15 |
10 |
5 |
2 |
|
|
Disease progression sites |
|
|
|
|
|
0.008 |
|
Others (n=53) |
16 |
13 |
9 |
9 |
6 |
|
|
Visceral meta (n=64) |
20 |
14 |
20 |
4 |
6 |
|
|
Bone only (n=47) |
11 |
18 |
2 |
10 |
6 |
|
|
PFS of the first line treatment |
|
|
|
|
|
0.209 |
|
ER driven (n=98) |
25 |
28 |
15 |
16 |
14 |
|
|
Not driven(n=66) |
22 |
17 |
16 |
7 |
4 |
|
1: years of age; 2: performance status; 3: Estrogen receptor; 4: Progesterone receptor; 5:diseae recurrence before 24 months of adjuvant endocrine therapy; 6: disease recurrence between after 24 months of adjuvant endocrine therapy and after 12 months of the end of endocrine therapy; 7: progression free survival
Lines 153-156 : “Other clinical factors including eastern cooperative oncology group(ECOG) performance status (p=0.001), initial visceral metastasis (p=0.006) and endocrine resistance in adjuvant setting (p=0.011) were associated with second-line treatment strategies.”
- In the part about “Factors for OS2” in Table 2 and in Table 4, please provide PFS2 as a factor.
: PFS2 is continuous factor not numeric factor. So, we made PFS2 to categorical factors according to median PFS2(months). In addition, ECOG performance status, initial visceral metastasis and endocrine resistance state were associated to 2nd line treatment and therefore we added these factors for revising table 2.
Table 2. Clinical characteristics that affected progression-free survival 2 and overall survival 2 (N=164)
|
Factors for PFS2 |
Ref |
N |
Hazard ratio |
95% CI |
|
P-value |
|
ECOG PS1 |
0 |
98 |
|
|
|
0.793 |
|
1-2 |
|
66 |
1.054 |
0.710 |
1.566 |
|
|
Endocrine resistance |
De novo |
70 |
|
|
|
0.778 |
|
Primary resistance |
|
28 |
1.000 |
0.603 |
1.660 |
|
|
Secondary resistance |
|
27 |
0.814 |
0.488 |
1.358 |
|
|
No resistance |
|
39 |
0.817 |
0.511 |
1.307 |
|
|
Initial visceral metastasis |
No |
120 |
|
|
|
0.596 |
|
Yes |
|
44 |
1.166 |
0.717 |
1.896 |
|
|
ER2-driven BC3 |
|
66 |
|
|
|
0.078 |
|
Yes |
No |
98 |
0.725 |
0.507 |
1.036 |
|
|
Disease progression site |
|
53 |
|
|
|
0.039 |
|
Visceral organ |
Other |
64 |
1.169 |
0.773 |
1.768 |
|
|
Bone only |
|
47 |
0.633 |
0.397 |
1.000 |
|
|
Second-line treatment |
Capecitabine |
47 |
|
|
|
0.031 |
|
Everolimus/exemestane |
|
45 |
1.665 |
1.038 |
2.670 |
|
|
Other cytotoxic chemo |
|
31 |
1.655 |
0.995 |
2.753 |
|
|
Fulvestrant |
|
23 |
2.383 |
1.364 |
4.163 |
|
|
Clinical trials |
|
18 |
1.713 |
0.904 |
3.246 |
|
|
Factors for OS2 |
Ref |
N |
Hazard ratio |
95% CI |
|
P-value |
|
ECOG PS1 |
0 |
98 |
|
|
|
0.720 |
|
1-2 |
|
66 |
0.897 |
0.494 |
1.628 |
|
|
Endocrine resistance |
De novo |
70 |
|
|
|
0.759 |
|
Primary resistance |
|
28 |
0.770 |
0.373 |
1.593 |
|
|
Secondary resistance |
|
27 |
0.687 |
0.297 |
1.590 |
|
|
No resistance |
|
39 |
0.980 |
0.465 |
2.065 |
|
|
Initial visceral metastasis |
No |
120 |
|
|
|
0.039 |
|
Yes |
|
44 |
2.097 |
1.039 |
4.234 |
|
|
ER2-driven BC3 |
No |
66 |
|
|
|
0.019 |
|
Yes |
|
98 |
0.525 |
0.306 |
0.901 |
|
|
Disease progression site |
Other |
53 |
|
|
|
0.026 |
|
Visceral organ |
|
64 |
2.339 |
1.166 |
4,234 |
|
|
Bone only |
|
47 |
0.967 |
0.461 |
2.027 |
|
|
Second-line treatment |
Capecitabine |
47 |
|
|
|
0.316 |
|
Everolimus/exemestane |
|
45 |
0.890 |
0.421 |
1.878 |
|
|
Other cytotoxic chemo |
|
31 |
1.253 |
0.600 |
2.617 |
|
|
Fulvestrant |
|
23 |
0.669 |
0.274 |
1.631 |
|
|
Clinical trials |
|
18 |
0.292 |
0.064 |
1.328 |
|
|
PFS24 |
≦5.2months |
47 |
|
|
|
<0.001 |
|
>5.2 months |
|
76 |
0.323 |
0.188 |
0.589 |
|
1: Eastern Cooperative Oncology Group Performance Status; 2: Estrogen receptor; 3: Breast cancer; 4:Progression free survival 2
Lines 175-179 : “In terms of OS2, PFS2 was the strongest prognostic factor of OS2 (hazard ratio:0.32, 95%Cis:0.19,0.59; p<0.001). In addition, ER-driven BC was associated with good OS2 (hazard ratio: 0.53, 95%CIs: 0.31, 0.90; p=0.019) whereas initial visceral metastasis and visceral organ disease progression were associated with poor OS2 (hazard ratio: 2.10, 95%CIs: 1.04, 4.23; p=0.039 and hazard ratio: 2.34, 95%CIs: 1.17, 4.23; p=0.026).”
For revising Table 4, we added PFS2 for multivariate analysis.
Table 4. Clinical characteristics that affected the OS in patients receiving second-line treatment (N=164)
|
Factors for OS |
Ref |
N |
Hazard ratio |
95% CI |
|
P-value |
|
Visceral metastasis |
No |
120 |
|
|
|
0.062 |
|
Yes |
|
44 |
1.935 |
0.968 |
3.867 |
|
|
Initial CA-15-3 |
Normal |
81 |
|
|
|
0.609 |
|
Elevation |
|
74 |
1.295 |
0.762 |
2.200 |
|
|
Unknown |
|
9 |
0.924 |
0.208 |
4.095 |
|
|
Germline BRCA status |
Wild type |
54 |
|
|
|
0.604 |
|
Mutation |
|
6 |
1.571 |
0.475 |
5.199 |
|
|
Not tested |
|
104 |
0.899 |
0.482 |
1.674 |
|
|
Number of meta organs |
1 |
74 |
|
|
|
0.048 |
|
2 or more |
|
90 |
1.705 |
1.004 |
2.894 |
|
|
ER-driven BC |
No |
66 |
|
|
|
<0.001 |
|
Yes |
|
98 |
0.180 |
0.103 |
0.316 |
|
|
Disease progression site |
Other |
53 |
|
|
|
0.034 |
|
Visceral organ |
|
64 |
2.191 |
1.099 |
4.368 |
|
|
Bone only |
|
47 |
0.932 |
0.444 |
1.956 |
|
|
Second-line treatment |
Capecitabine |
47 |
|
|
|
0.373 |
|
Everolimus/exemestane |
|
45 |
0.984 |
0.469 |
2.063 |
|
|
Other cytotoxic chemo |
|
31 |
1.318 |
0.624 |
2.786 |
|
|
Fulvestrant |
|
23 |
0.829 |
0.353 |
1.947 |
|
|
Clinical trials |
|
18 |
0.292 |
0.065 |
1.318 |
|
|
PFS21 |
≦5.2months |
47 |
|
|
|
<0.001 |
|
>5.2 months |
|
76 |
0.340 |
0.194 |
0.598 |
|
1: Progression Free Survival 2
In this analysis, PFS2 is significantly associated to OS and OS2. These contents were added in result section and discussion section.
- In Table 3, please provide the characteristics that were selected in Supplementary Table 1, information on the duration of the first-line therapy (or ER-driven tumors), and PFS2 as factors.
: We demonstrated that the impact of baseline characteristics on overall survival in Table 3. Therefore, ER-driven tumor and PFS2 did not exist in Table 3 but in Table 4. According to your comment, we revised table 3 and result section.
Table 3. Clinical characteristics that affected overall survival (N=305)
|
Factors |
Ref |
N |
Hazard ratio |
95% CI |
|
P-value |
|
Age |
<50 |
130 |
|
|
|
0.131 |
|
>50 years old |
|
175 |
1.493 |
0.888 |
2.510 |
|
|
ECOG PS1 |
0 |
177 |
|
|
|
0.658 |
|
1 |
|
122 |
0.793 |
0.483 |
1.301 |
|
|
2 |
|
5 |
1.543 |
0.434 |
5.484 |
|
|
Unknown |
|
1 |
- |
- |
- |
|
|
Visceral metastasis |
No |
239 |
|
|
|
0.068 |
|
Yes |
|
66 |
1.599 |
0.967 |
2.645 |
|
|
Initial CA-15-3 |
Normal |
272 |
|
|
|
0.019 |
|
Elevation |
|
31 |
1.922 |
1.184 |
3.120 |
|
|
Unknown |
|
2 |
0.706 |
0.67 |
2.986 |
|
|
Initial CEA |
Normal |
272 |
|
|
|
0.908 |
|
Elevation |
|
31 |
0.987 |
0.550 |
1.771 |
|
|
Unknown |
|
2 |
1.505 |
0.231 |
9.790 |
|
|
Endocrine resistance |
De novo |
70 |
|
|
|
0.031 |
|
Primary resistance |
|
28 |
2.250 |
1.171 |
4.323 |
|
|
Secondary resistance |
|
27 |
0.854 |
0.427 |
1.705 |
|
|
No resistance |
|
39 |
0.898 |
0.499 |
1.578 |
|
|
Germline BRCA status |
Normal |
92 |
|
|
|
0.040 |
|
Mutation |
|
6 |
3.989 |
1.311 |
12.139 |
|
|
Not tested |
|
207 |
1.460 |
0.867 |
2.459 |
|
|
Number of meta organs |
1 |
157 |
|
|
|
0.021 |
|
2 or more |
|
148 |
1.774 |
1.091 |
2.886 |
|
1: Eastern Cooperative Oncology Group Performance Status
Lines 222 – 227 : “The clinical factors that affected the OS were analyzed in all 305 patients (Table 3). In this analysis, visceral metastasis (hazard ratio: 1.60, 95% CIs: 0.97, 2.65; p=0.068), initial CA-15-3 elevation (hazard ratio: 1.92, 95% CIs: 1.18, 3.12), endocrine resistance (hazard ratio for primary resistance: 2.25, 95% CIs: 1.17,4.32), number of metastatic organs (hazard ratio: 1.774, 95% CIs: 1.091, 2.886) and germline BRCA mutation (hazard ratio: 3.99, 95% CIs: 1.31, 12.14) were all associated with a short duration of OS (p<0.05, respectively).
- Please show a summary of the second-line treatment strategy, including the factors that the authors recommend should be considered after palbociclib and letrozole, in a figure in the Discussion section.
: Thanks for your suggestion. We presented the summary diagram as your comment.

Minor comments:
- Line 28 and line 33 in the Abstract; The full names for “PFS2” and “OS2” should be provided.
: Thanks for your comment. We revised our manuscript according to your comment.
Line 26 : “the median progression free survival of 2nd line treatment in metastatic setting (PFS2)”
Line 31 : “The median overall survival of 2nd line treatment in metastatic setting (OS2)”
- Line 52; “And” should be deleted. 3. Line 289; “…. in patients treated with AI alone as the first-line treatment.” Is the term “alone” necessary?
: Thanks for your kind review. We removed these typos according to your comment.